# Unsupervised Learning of Entailment-Vector Word Embeddings

## Abstract

Entailment vectors are a principled way to encode in a vector what information is known and what is unknown. They are designed to model relations where one vector should include all the information in another vector, called entailment. This paper investigates the unsupervised learning of entailment vectors for the semantics of words. Using simple entailment-based models of the semantics of words in text (distributional semantics), we induce entailment-vector word embeddings which outperform the best previous results for predicting entailment between words, in unsupervised and semi-supervised experiments on hyponymy.

## 1 Introduction

Modelling entailment, is a fundamental issue in the semantics of natural language, and there has been a lot of interest in modelling entailment using vector-space representations. But, until recently, unsupervised models such as word embeddings have performed surprisingly poorly at detecting entailment Weeds et al. (2014); Shwartz et al. (2017), not beating a frequency baseline Weeds et al. (2014). Entailment is the relation of information inclusion, meaning that $y$ entails $x$ if and only if everything that is known given $x$ is also known given $y$. As such, representations which support entailment need to encode not just what information is known, but also what information is unknown. The results on lexical entailment seem to indicate that standard word embeddings, such as Word2Vec, do not reflect the relative abstractness of words, and in this sense do not reflect how much information is left unspecified by a word.

In contrast with the majority of the work in this area, which simply uses existing vector-space embeddings of words in their models of entailment, recent work has addressed this issue by proposing new vector-space models which are specifically designed to capture entailment. In particular, Vilnis & McCallum (2015) use variances to represent the uncertainty in values in a continuous space, and Henderson & Popa (2016) use probabilities to represent uncertainty about a discrete space. We will refer to the latter as the "entailment-vectors" framework. In this work, we use this framework from Henderson & Popa (2016) to develop new entailment-based models for the unsupervised learning of word embeddings, and demonstrate that these embeddings achieve unprecedented results in predicting entailment between words.

Our unsupervised models use the distribution of words in a large text corpus to induce vector-space representations of the meaning of words. This approach to word meaning is called distributional semantics. The distributional semantic hypothesis (Harris, 1954) says that the meaning of a word is reflected in the distribution of text contexts which it appears in. Many methods (e.g. (Deerwester et al., 1990; Schütze, 1993; Mikolov et al., 2013a) and this paper) have been proposed for inducing vector representations of the meaning of words (word embeddings) from the distribution of word-context pairs found in large corpora of text.

In the framework of Henderson & Popa (2016), each dimension of the vector-space represents something that might be known, and continuous vectors represent probabilities of these features being known or unknown. Henderson & Popa (2016) illustrate their framework by proposing a reinterpretation of existing Word2Vec (Mikolov et al., 2013a) word embeddings which maps them into entailment vectors, which in turn successfully predict entailment between words (hyponymy). To motivate this reinterpretation of existing word embeddings, they propose a model of distributional semantics and argue that the Word2Vec training objective approximates the training objective of this distributional semantic model given the mapping.

In this paper, we implement this distributional semantic model and train new word embeddings using the exact objective. Based on our analysis of this model, we propose that this implementation can be done in several ways, including the one which motivates Henderson & Popa (2016)'s reinterpretation of Word2Vec embeddings. In each case, training results in entailment vector embeddings, which directly encode what is known and unknown given a word, and thus do not require any reinterpretation to predict hyponymy.

To model the semantic relationship between a word and its context, the distributional semantic model postulates a latent pseudo-phrase vector for the unified semantics of the word and its neighbouring context word. This latent vector must entail the features in both words' vectors and must be consistent with a prior over semantic vectors, thereby modelling the redundancy and consistency between the semantics of two neighbouring words.

Based on our analysis of this entailment-based distributional semantic model, we hypothesise that the word embeddings suggested by Henderson & Popa (2016) are in fact not the best way to extract information about the semantics of a word from this model. They propose using a vector which represents the evidence about known features given the word (henceforth called the likelihood vectors). We propose to instead use a vector which represents the posterior distribution of known features for a phrase containing only the word. This posterior vector includes both the evidence from the word and its indirect consequences via the constraints imposed by the prior. Our efficient implementation of this model allows us to test this hypothesis by outputting either the likelihood vectors or the posterior vectors as word embeddings.

To evaluate these word embeddings, we predict hyponymy between words, in both an unsupervised and semi-supervised setting. Given the word embeddings for two words, we measure whether they are a hypernym-hyponym pair using an entailment operator from (Henderson & Popa, 2016) applied to the two embeddings. We find that using the likelihood vectors performs as well as reinterpreting Word2Vec embeddings, confirming the claims of equivalence by Henderson & Popa (2016). But we also find that using the posterior vectors performs significantly better, confirming our hypothesis that posterior vectors are better, and achieving the best published results on this benchmark dataset. In addition to these unsupervised experiments, we evaluate in a semi-supervised setting and find a similar pattern of results, again achieving state-of-the-art performance.

In the rest of this paper, section 2 presents the formal framework we use for modelling entailment in a vector space, the distributional semantic models, and how these are used to predict hyponymy. Section 3 discusses additional related work, and then section 4 presents the empirical evaluation on hyponymy detection, in both unsupervised and semi-supervised experiments. Some additional analysis of the induced vectors is presented in section 4.4.

## 2 DISTRIBUTIONAL SEMANTIC ENTAILMENT

Distributional semantics uses the distribution of contexts in which a word occurs to induce the semantics of the word (Harris, 1954; Deerwester et al., 1990; Schütze, 1993). The Word2Vec model (Mikolov et al., 2013a) introduced a set of refinements and computational optimisations of this idea which allowed the learning of vector-space embeddings for words from very large corpora with very good semantic generalisation. Henderson & Popa (2016) motivate their reinterpretation the Word2Vec Skipgram (Mikolov et al., 2013a) distributional semantic model with an entailment-based model of the semantic relationship between a word and its context words. We start by explaining our interpretation of the distributional semantic model proposed by Henderson & Popa (2016), and then propose our alternative models.

Henderson & Popa (2016) postulate a latent vector $y$ which is the consistent unification of the features of the middle word $x'_e$ and the neighbouring context word $x_e$, illustrated on the left in figure 1.[1] We can think of the latent vector $y$ as representing the semantics of a pseudo-phrase consisting of the two words. The unification requirement is defined as requiring that $y$ entail both words, written $y{\Rightarrow}x'_e$ and $y{\Rightarrow}x_e$. The consistency requirement is defined as $y$ satisfying a prior $\theta(y)$, which embodies all the the constraints and correlations between features in the vector. This approach models the relationship between the semantics of a word and its context as being redundant and consistent.

---

[1]Note that "$x_e$" is being used here as the name of a whole vector, not to be confused with "$x_i$", which refers to element $i$ in vector $x$.

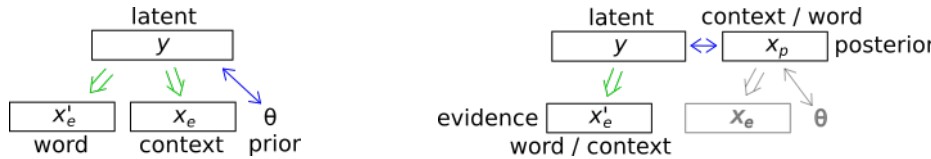

Figure 1: The distributional semantic model of a word and its context (left), and its approximation in the word2hyp models (right).

If $x'_e$ and $x_e$ share features, then it will be easier for $y$ to satisfy both $y{\Rightarrow}x'_e$ and $y{\Rightarrow}x_e$. If the features of $x'_e$ and $x_e$ are consistent, then it will be easier for $y$ to satisfy the prior $\theta(y)$.

## 2.1 THE REINTERPRETATION OF WORD2VEC

Henderson & Popa (2016) formalise the above model using their entailment-vectors framework. This framework models distributions over discrete vectors where a 1 in position $i$ means feature $i$ is known and a 0 means it is unknown. Entailment $y{\Rightarrow}x$ requires that the 1s in $x$ are a subset of the 1s in $y$, so $1{\Rightarrow}1$, $0{\Rightarrow}0$ and $1{\Rightarrow}0$, but $0{\not\Rightarrow}1$. Distributions over these discrete vectors are represented as continuous vectors of log-odds $X$, so $P(x_i{=}1) = \sigma(X_i)$, where $\sigma$ is the logistic sigmoid. The probability of entailment $y{\Rightarrow}x$ between two such "entailment vectors" $Y, X$ can be measured using the operator $\oslash$:[2]

$$\begin{aligned} \log P(y{\Rightarrow}x \mid Y, X) &\approx \\ Y{\oslash}X &\equiv \sigma(-Y) \cdot \log \sigma(-X) \end{aligned} \tag{1}$$

For each feature $i$ in the vector, it calculates the expectation according to $P(y_i)$ that, either $y_i{=}1$ and thus the log-probability is zero, or $y_i{=}0$ and thus the log-probability is $\log P(x_i{=}0)$ (noting that $\sigma(-X_i) = (1 - \sigma(X_i)) \approx P(x_i{=}0)$).

Henderson & Popa (2016) formalise the model on the left in figure 1 by first inferring the optimal latent vector distribution $Y$ (equation (3)), and then scoring how well the entailment and prior constraints have been satisfied (equation (2)).

$$\begin{aligned} \max_Y (E_{Y,X'_e,X_e} \log P(y{\Rightarrow}x'_e, \, y{\Rightarrow}x_e, \, y)) & \\ \approx Y{\oslash}X'_e + Y{\oslash}X_e + (-\sigma(-Y)) \cdot \theta(Y) & \end{aligned} \tag{2}$$

where

$$Y = -\log \sigma(-X'_e) + -\log \sigma(-X_e) + \theta(Y) \tag{3}$$

where $E_{Y,X'_e,X_e}$ is the expectation over the distribution defined by the log-odds vectors $Y, X'_e, X_e$, and $\log$ and $\sigma$ are applied componentwise. The term $\theta(Y)$ is used to indicate the net effect of the prior on the vector $Y$. Note that, in the formula (3) for inferring $Y$, the contribution $-\log \sigma(-X)$ of each word vector is also a component of the definition of $Y{\oslash}X$ from equation (1). In this way, the score for measuring how well the entailment has been satisfied is using the same approximation as used in the inference to satisfy the entailment constraint. This function $-\log \sigma(-X)$ is a non-negative transform of $X$, as shown in figure 2. Intuitively, for an entailed vector $x$, we only care about the probability that $x_i{=}1$ (positive log-odds $X_i$), because that constrains the entailing vector $y$ to have $y_i{=}1$ (adding to the log-odds $Y_i$).

The above model cannot be mapped directly to the Word2Vec model because Word2Vec has no way to model the prior $\theta(Y)$. On the other hand, the Word2Vec model postulates two vectors for every word, compared to one in the above model. Henderson & Popa (2016) propose an approximation to the above model which incorporates the prior into one of the two vectors, resulting in each word having one vector $X_e$ as above plus another vector $X_p$ with the prior incorporated.

$$X_p \approx -\log \sigma(-X_e) + \theta(Y) \tag{4}$$

---

[2]We use lowercase variables $x, y$ to refer to discrete vectors and uppercase variables $X, Y$ to refer to their associated entailment vectors. Entailment vectors represent the factorised distributions which result from a variational Bayesian approximation. In this paper we only review the results of this approximation which are necessary for our models, including this scoring function (1) and the following inference formula (3).

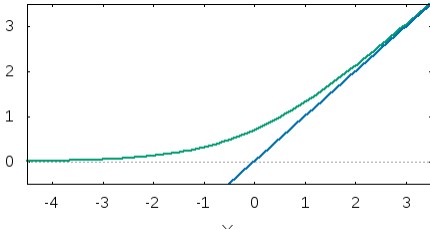

Figure 2: The function $-\log \sigma(-X)$ used in inference and the $\oslash$ operator, versus $X$.

Both vectors $X_e$ and $X_p$ are parameters of the model, which need to be learned. Thus, there is no need to explicitly model the prior, thereby avoiding the need to choose a particular form for the prior $\theta$, which in general may be very complex.

This gives us the following score for how well the constraints of this model can be satisfied.

$$\max_Y (E_{Y,X'_e,X_p} \log P(y{\Rightarrow}x'_e,\, y{\Rightarrow}x_e,\, y))$$
$$\approx Y \oslash X'_e + (-\sigma(-Y)) \cdot X_p \tag{5}$$
where
$$Y = -\log \sigma(-X'_e) + X_p \tag{6}$$

In (Henderson & Popa, 2016), score (5) is only used to provide a reinterpretation of Word2Vec word embeddings. They show that a transformation of the vectors output by Word2Vec ("W2V *u.d.*$\oslash$" below) can be seen as an approximation to the likelihood vector $X_e$. In Section 4, we empirically test this hypothesis by directly training $X_e$ ("W2H likelihood" below) and comparing the results to those with reinterpreted Word2Vec vectors.

## 2.2 New Distributional Semantic Models

In this paper, we implement distributional semantic models based on score (5) and use them to train new word embeddings. We call these models the Word2Hyp models, because they are based on Word2Vec but are designed to predict hyponymy.

To motivate our models, we provide a better understanding of the model behind score (5). In particular, we note that although we want $X_p$ to approximate the effects $\theta(Y)$ of the prior as in equation 4, in fact $X_p$ is only dependent on one of the two words, and thus can only incorporate the portion of $\theta(Y)$ which arises from that one word. Thus, a better understanding of $X_p$ is provided by equation (7).

$$X_p \approx -\log \sigma(-X_e) + \theta(X_p) \tag{7}$$

In this framework, equation (7) is exactly the same formula as would be used to infer the vector for a single-word phrase (analogously to equation (3)).

This interpretation of the approximate model in equation 5 is given on the right side of figure 1. As shown, $X_p$ is interpreted as the *posterior* vector for a single-word phrase, which incorporates the likelihood and the prior for that word. In contrast, $X'_e$ is just the *likelihood*, which provides the evidence about the features of $Y$ from the other word, without including the indirect consequences of this information. This model, as argued above, approximates the model on the left side in Figure 1. But the grey part of the figure does not need to be explicitly modelled because $X_p$ is trained directly.

This interpretation suggests that the posterior vector $X_p$ should be a better reflection of the semantics of the word than the likelihood vector $X_e$, since it includes both the direct evidence for some features and their indirect consequences for other features. We test this hypothesis empirically in Section 4.

To implement our distributional semantic models, we define new versions of the Word2Vec code (Mikolov et al., 2013a;b). The Word2Vec code trains two vectors for each word, where negative sampling is applied to one of these vectors, and the other is the output vector. This applies to both the Skipgram and CBOW versions of training. Both versions also use a dot product between vectors to try to predict whether the example is a positive or negative sample. We simply replace this dot

product with score (5) directly in the Word2Vec code, leaving the rest of the algorithm unchanged. We make this change in one of two ways, one where the output vector corresponds to the likelihood vector $X_e$, and one where the output vector corresponds to the posterior vector $X_p$. We will refer to the model where $X_p$ is output as the "posterior" model, and the model where $X_e$ is output as the "likelihood" model. Both these methods can be applied to both the Skipgram and CBOW models, giving us four different models to evaluate.

### 2.3 MODELLING HYPONYMY

The proposed distributional semantic models output a word embedding vector for every word in the vocabulary, which are directly interpretable as entailment vectors in the entailment-vectors framework. Thus, to predict lexical entailment between two words, we can simply apply the $\oslash$ operator to their vectors, to get an approximation of the log-probability of entailment.

We evaluate these entailment predictions on hyponymy detection. Hyponym-hypernym pairs should have associated embeddings $Y, X$ which have a higher entailment scores $Y \oslash X$ than other pairs. We rank the word pairs by the entailment scores for their embeddings, and evaluate this ranked list against the gold hyponymy annotations. We evaluate on hyponymy detection because it reflects a direct form of lexical entailment; the semantic features of a hypernym (e.g. "animal") should be included in the semantic features of the hyponym (e.g. "cat"). Other forms of lexical entailment would benefit from some kind of reasoning or world knowledge, which we leave to future work on compositional models.

## 3 RELATED WORK

In this paper we propose a distributional semantic model which is based on entailment. Most of the work on modelling entailment with vector space embeddings has simply used distributional semantic vectors within a model of entailment, and is therefore not directly relevant here. See (Shwartz et al., 2017) for a comprehensive review of such measures. Shwartz et al. (2017) evaluate these measures as unsupervised models of hyponymy detection and run experiments on a number of hyponymy datasets. We report their best comparable result in Table 1.

Vilnis & McCallum (2015) propose an unsupervised model of entailment in a vector space, and evaluate it on hyponymy detection. Instead of representing words as a point in a vector space, they represent words as a Gaussian distribution over points in a vector space. The variance of this distribution in a given dimension indicates the extent to which the dimension's feature is unknown, so they use KL-divergence to detect hyponymy relations. Although this model has a nice theoretical motivation, the word representations are more complex and training appears to be more computationally expensive than the method proposed here. We empirically compare our models to their hyponymy detection accuracy and find equivalent results.

The semi-supervised model of Kruszewski et al. (2015) learns a discrete Boolean vector space for predicting hyponymy. But they do not propose any unsupervised method for learning these vectors.

Weeds et al. (2014) report hyponymy detection results for a number of unsupervised and semi-supervised models. They propose a semi-supervised evaluation methodology where the words in the training and test sets are disjoint, so that the supervised component must learn about the unsupervised vector space and not about the individual words. Following Henderson & Popa (2016), we replicate their experimental setup in our evaluations, for both unsupervised and semi-supervised models, and compare to the best results among the models evaluated by Weeds et al. (2014), Shwartz et al. (2017) and Henderson & Popa (2016).

## 4 EVALUATION OF WORD EMBEDDINGS

We evaluate on hyponymy detection in both a fully unsupervised setup and a semi-supervised setup. In the semi-supervised setup, we use labelled hyponymy data to train a linear mapping from the unsupervised vector space to a new vector space with the objective of correctly predicting hyponymy relations in the new vector space. This prediction is done with the same (or equivalent) entailment operator as for the unsupervised experiments (called "map $\oslash$" in Table 2).

| embeddings | operator | | 50% Acc | Ave Prec |
|---|---|---|---|---|
| Weeds et.al., 2014 | | | 58% | – |
| Shwartz et.al., 2017 | | | – | 44.1% |
| W2V GoogleNews | u.d.⊘ | | 64.5%* | – |
| W2V CBOW | u.d. ⊘ | | 53.2% | 55.2% |
| W2H Skip | likelihood | ⊘ | 59.5% | 57.8% |
| W2H CBOW | likelihood | ⊘ | 61.8% | 66.4% |
| W2V Skip | u.d.⊘ | | 62.1% | 67.6% |
| W2H CBOW | posterior | ⊘ | 68.1%* | **70.8%** |
| W2H Skip | posterior | ⊘ | **69.6%** | 68.9% |

Table 1: Hyponymy detection accuracies (*50% Acc*) and average precision (*Ave Prec*), in the unsupervised experiments. For the accuracies, * marks a significant improvement over the higher rows.

We replicate the experimental setup of Weeds et al. (2014), using their selection of hyponym-hypernym pairs from the BLESS dataset (Baroni & Lenci, 2011), which consists of noun-noun pairs, including 50% positive hyponymy pairs plus 50% negative pairs consisting of some other hyponymy pairs reversed, some pairs in other semantic relations, and some random pairs. As in (Weeds et al., 2014), our semi-supervised experiments use ten-fold cross validation, where each fold has items removed from the training set if they contain a word that also occurs in the testing set.

The word embedding vectors which we train have 200 dimensions and were trained using our Word2Hyp modification of the Word2Vec code (with default settings), trained on a corpus of half a billion words of Wikipedia. We also replicate the approach of Henderson & Popa (2016) by training Word2Vec embeddings on this data.

To quantify performance on hyponymy detection, for each model we rank the list of pairs according to the score given by the model, and report two measures of performance for this ranked lists. The "*50% Acc*" measure treats the first half of the list as labelled positive and the second half as labelled negative. This is motivated by the fact that we know a priori that the proportion of positive examples has been artificially set to (approximately) 50%. Average precision is a measure of the accuracy for ranked lists, used in Information Retrieval and advocated as a measure of hyponymy detection by Vilnis & McCallum (2015). For each positive example, precision is measured at the threshold just below that example, and these precision scores are averaged over positive examples. For cross validation, we average over the union of positive examples in all the test sets. Both these measures are reported (when available) in Tables 1 and 2.

## 4.1 Unsupervised Hyponymy Detection

The first set of experiments evaluate the different embeddings in their unsupervised models of hyponymy detection. Results are shown in Table 1. Our principal point of comparison is the best results from (Henderson & Popa, 2016) (called "W2V GoogleNews" in Table 1). They use the pre-existing publicly available GoogleNews word embeddings, which were trained with the Word2Vec software on 100 billion words of the GoogleNews dataset, and have 300 dimensions. To provide a more direct comparison, we replicate the model of Henderson & Popa (2016) but using the same embedding training setup as for our Word2Hyp model ("W2V Skip"). Both cases use their proposed reinterpretation of these vectors for predicting entailment ("u.d.⊘"). We also report the best results from Weeds et al. (2014) and the best comparable results from (Shwartz et al., 2017). For our proposed Word2Hyp distributional semantic models ("W2H"), we report results for the four combinations of using the CBOW or Skipgram ("Skip") model to train the likelihood or posterior vectors.

The two Word2Hyp models with likelihood vectors perform slightly better than the best unsupervised model of Weeds et al. (2014), but similarly. The reinterpretation of Word2Vec vectors ("W2V GoogleNews u.d.⊘") performs significantly better, but when the same method is applied to the smaller Wikipedia corpus ("W2V Skip u.d.⊘"), this difference all but disappears. This confirms the hypothesis of Henderson & Popa (2016) that the reinterpretation of Word2Vec vectors and the likelihood vectors from Word2Hyp are approximately equivalent.

| embeddings | operator | 50% Acc | Ave Prec |
|---|---|---|---|
| Weeds et.al., 2014 | | 75% | – |
| W2V GoogleNews | *map* ⊘ | 80.1% | – |
| W2V Skip | *map* ⊘ | 81.9% | 88.3% |
| W2H CBOW likelihood | *map* ⊘ | 83.3% | 90.3% |
| W2V CBOW | *map* ⊘ | 84.6% | 91.5% |
| W2H Skip likelihood | *map* ⊘ | 84.8% | 90.9% |
| W2H Skip posterior | *map* ⊘ | 85.5% | 91.3% |
| W2H CBOW posterior | *map* ⊘ | **86.0%** | **92.8%** |

Table 2: Hyponymy detection accuracies (*50% Acc*) and average precision (*Ave Prec*), in the semi-supervised experiments.

However, even with this smaller corpus, using the proposed posterior vectors from the Word2Hyp model are significantly more accurate than the reinterpretation of Word2Vec vectors. This confirms the hypothesis that the posterior vectors from the Word2Hyp model are a better model of the semantics of a word than the likelihood vectors suggested by Henderson & Popa (2016).

Using the CBOW model or the Skipgram model makes only a small difference. The average precision score shows the same pattern as the accuracy.

To allow a direct comparison to the model of Vilnis & McCallum (2015), we also evaluated the unsupervised models on the hyponymy data from (Baroni et al., 2012), which is not as carefully designed to evaluate hyponymy as the (Weeds et al., 2014) data. Both the likelihood and posterior vectors of the Word2Hyp CBOW model achieved average precision (81%, 80%) which is not significantly different from the best model of Vilnis & McCallum (2015) (80%).

## 4.2 SEMI-SUPERVISED HYPONYMY DETECTION

The semi-supervised experiments train a linear mapping from each unsupervised vector space to a new vector space, where the entailment operator ⊘ is used to predict hyponymy ("map ⊘").

The semi-supervised results (shown in Table 2)[3] no longer show an advantage of GoogleNews vectors over Wikipedia vectors for the reinterpretation of Word2Vec vectors. And the advantage of posterior vectors over the likelihood vectors is less pronounced. However, the two posterior vectors still perform much better than all the previously proposed models, achieving 86% accuracy and nearly 93% average precision. These semi-supervised results confirm the results from the unsupervised experiments, that Word2Vec embeddings and Word2Hyp likelihood embeddings perform similarly, but that using the posterior vectors of the Word2Hyp model perform better.

## 4.3 TRAINING TIMES

Because the similarity measure in equation 5 is more complex than a simple dot product, training a new distributional semantic model is slower than with the original Word2Vec code. In our experiments, training took about 8 times longer for the CBOW model and about 15 times longer for the Skipgram model. This meant that Word2Hyp CBOW trained about 8 times faster than Word2Hyp Skipgram. As in the Word2Vec code, we used a quadrature approximation (i.e. a look-up table) to speed up the computation of the sigmoid function, and we added the same technique for computing the log-sigmoid function.

## 4.4 DISCUSSION

The relative success of our distributional semantic models at unsupervised hyponymy detection indicates that they are capturing some aspects of lexical entailment. But the gap between the unsupervised and semi-supervised results indicates that other features are also being captured. This is not surprising, since many other factors influence the co-occurrence statistics of words.

---

[3]It is not clear how to measure significance for cross-validation results, so we do not attempt to do so.

| most abstract | | least abstract | |
| --- | --- | --- | --- |
| *something* | *necessity* | *. . .* | *fork* |
| *anything* | *sense* | *hockey* | *housing* |
| *end* | *back* | *republican* | *elm* |
| *inside* | *saw* | *hull* | *primate* |
| *good* | *. . .* | *cricket* | *fur* |

Table 3: Ranking of the abstractness ($\mathbf{0} \oslash X$) of frequent words from the hyponymy dataset, using Word2Hyp-Skipgram-posterior embeddings.

To get a better understanding of these word embeddings, we ranked them by degree of abstractness. Table 3 shows the most abstract and least abstract frequent words that occur in the hyponymy data. To measure abstractness, we used our best unsupervised embeddings and measured how well they are entailed by the zero log-odds vector, which represents a uniform half probability of knowing each feature. For a vector to be entailed by the zero vector, it must be that its features are mostly probably unknown. The less you know given a word, the more abstract it is.

An initial ranking found that six of the top ten abstract words had frequency less than 300 in the Wikipedia data, but none of the ten least abstract terms were infrequent. This indicates a problem with the current method, since infrequent words are generally very specific (as was the case for these low-frequency words, *submissiveness, implementer, overdraft, ruminant, warplane,* and *londoner*). Although this is an interesting characteristic of the method, the terms themselves seem to be noise, so we rank only terms with frequency greater than 300.

The most abstract terms in table 3 include some clearly semantically abstract terms, in particular *something* and *anything* are ranked highest. Others may be affected by lexical ambiguity, since the model does not disambiguate words by part-of-speech (such as *end*, *good*, *sense*, *back*, and *saw*). The least abstract terms are mostly very semantically specific, but it is indicative that this list includes *primate*, which is an abstract term in Zoology but presumably occurs in very specific contexts in Wikipedia.

## 5 Conclusions

In this paper, we propose unsupervised methods for efficiently training word embeddings which capture semantic entailment. This work builds on the work of Henderson & Popa (2016), who propose the entailment-vectors framework for modelling entailment in a vector-space, and a distributional semantic model for reinterpreting Word2Vec word embeddings. Our contribution differs from theirs in that we provide a better understanding of their distributional semantic model, we choose different vectors in the model to use as word embeddings, and we train new word embeddings using our modification of the Word2Vec code. Empirical results on unsupervised and semi-supervised hyponymy detection confirm that the model's likelihood vectors, which Henderson & Popa (2016) suggest to use, do indeed perform equivalently to their reinterpretation of Word2Vec vectors. But these experiments also show that the model's posterior vectors, which we propose to use, perform significantly better, outperforming all previous results on this benchmark dataset.

The success of these unsupervised models demonstrates that the proposed distributional semantic models are effective at extracting information about lexical entailment from the redundancy and consistency of words with their contexts in large text corpora. The use of the entailment-vectors framework to efficiently model entailment relations has been crucial to this success. This result suggests future work using the entailment-vectors framework in unsupervised models that leverage other distributional evidence about semantics, particularly in models of compositional semantics. The merger of word embeddings with compositional semantics to get representation learning for larger units of text is currently an important challenge in the semantics of natural language, and the work presented in this paper makes a significant contribution towards solving it.

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
