# OpenReview forum: "Unsupervised Learning of Entailment-Vector Word Embeddings"
_ICLR.cc/2018/Conference — Reject_

### Official Review · AnonReviewer3 · 2017-11-22
**Model is unclear, evaluation needs expansion**

**Rating:** 3
**Confidence:** 5

**Review:**

I'm finding this paper really difficult to understand. The introduction is very abstract, and it is hard for me to understand the model as it is explained at the moment. Could the authors please clarify, perhaps in more algorithmic terms, how the model works?

As for the evaluation, BLESS is a nice dataset, but it certainly isn't enough to make a broad claim because it has certain artifacts in the way negative examples were constructed. I recommend looking at the collection of datasets used by Levy et al. [1] and Shwartz et al. [2], and evaluating on their union.

Another discrepancy that appears in the paper is that the authors cite Shwartz et al. [2] as achieving 44.1% average precision on BLESS, when in fact, this number reflects their performance on the WordNet-based dataset created by Weeds et al. [3].

[1] http://www.aclweb.org/anthology/N15-1098
[2] http://aclweb.org/anthology/E/E17/E17-1007.pdf
[3] http://sro.sussex.ac.uk/53103/1/C14-1212.pdf

---

> ### Author Response · Authors · 2017-12-22
> **Reviewer has misunderstood the evaluation**
>
> Reviewer has misunderstood the evaluation.  In fact, our tables of results are all on the dataset from Weeds et al. 2014, which is precisely designed to address the criticism Reviewer points out for the previous BLESS-based datasets.  We only report results on an earlier BLESS dataset to allow direct comparison to previous work.  Hence the reported result from Shwartz is the appropriate one.
>
> We can try to make the model easier to understand, but the fundamental difficulty comes from the novelty of the model, not from the writing.  If a reader is not familiar with Henderson and Popa 2016, then it is impossible to explain this vector-space model convincingly and still have space for the novel contributions of this submission.  This vector-space representation is just too different from previous vector-space representations.

---

### Official Review · AnonReviewer2 · 2017-11-25
**a word embedding algorithm for  lexical entailment. Good experimental results.**

**Rating:** 7
**Confidence:** 3

**Review:**

The paper presents a word embedding algorithm for lexical entailment. The paper follows the work of Henderson and Popa (ACL,2016) that presented an interpretation of word2vec word representation in which each feature in a word vector corresponds to the probability of it being known/unknown, and suggested an operator to compute the degree of entailment between two words. In this paper, the authors train the word2vec algorithms to directly optimize the objective function suggested by Henderson and Popa (2016),
I find the paper interesting. The proposed approach is novel and not standard and the paper reports significant improvement of entailment results  compared to previous state of the art

The method part of the paper (sections 2.1 and 2.2 ) which is the main contribution is not clearly written. The paper heavily relies on Henderson and Popa (2016). You dont need to restate in the current paper all the mathematical analysis that appears in the previous paper. You are expected, however, that the model description and the notation that is used here should be clearly explained. Maybe you can also add algorithm box. I think that the author should prepared a revised version of section 2.

In Word2vec, Levy and Goldberg provided an elegant analysis of the algorithm and showed that the global optimum is obtained at the PMI matrix. Can you derive a similar analysis for your variant of the word2vec algorithm?

---

> ### Author Response · Authors · 2017-12-22
> **Thanks for the suggestions**
>
> We thank Reviewer for the suggestions, which we will try to take into account.  We can understand the reviewers' desire to make everything a simple extension of something we already understand, but it is not possible in this case.
>
> Unlike for Levy and Goldberg's insightful analysis of word2vec, there is no equivalence between our model and a previously proposed model, other than the connections to Henderson and Popa 2016 stated in the submission.
>
> All three reviewers found it hard to understand the model as it is presented in section 2.  We can try to rewrite section 2 again, but the fundamental difficulty comes from the novelty of the model, not from the writing.  If a reader is not familiar with Henderson and Popa 2016, then it is impossible to explain this vector-space model convincingly and still have space for the novel contributions of this submission.  This vector-space representation is just too different from previous vector-space representations.

---

### Official Review · AnonReviewer1 · 2017-11-28

**Rating:** 3
**Confidence:** 5

**Review:**

This work proposes to learn word vectors that are intended to specifically model the lexical entailment relationship. This is achieved in an unsupervised manner from unstructured data, through an approach heavily influenced by recent work by Henderson and Popa, which "reinterprets word2vec" by modeling distributions over discrete latent "pseudo-phrase" vectors. That is, instead of using two vectors per word, as in word2vec, a latent representation is introduced that models the joint properties of the target and context words. While Henderson and Popa represent the latent vector as the evidence for the target and context, or the likelihood, this work suggests to represent it based on the posterior distribution instead. The resultant representations are evaluated on Weeds et al.'s (2014) version of BLESS, as well as the full BLESS dataset, where they do better than the original.

The paper is confusingly written, fails to mention a lot of related work, has a weak evaluation where it doesn't compare to related systems, and I feel that it would benefit from "toning down". Hence, I do not recommend it for acceptance. In more detail:

1. The idea behind Henderson and Popa's model, as well as the suggested modification, should be easy to explain, but I really had to struggle to make sense of it. This work relies very heavily on that paper, and would be better off if it was more standalone. I think part of the confusion stems from using y for the latent representation but not specifying whether it is a word or latent representation in Equation 1 - that only becomes obvious later. The exposition clearly needs more work, and more precise technical writing.

2. There is a lot of related work around word embeddings that is not mentioned, both on word2vec-style representation learning (e.g. it would be useful to relate this more to word2vec and what it learns, as in Omer Levy's work on "interpreting" word2vec, rather than reinterpreting) and word embeddings on hypernymy detection and lexical entailment (see e.g. Stephen Roller's thesis for references).

3. There has been a lot of work on the Weeds BLESS dataset that is not mentioned, or compared against, including unsupervised approaches (e.g. Levy's work, Santus's work, Kiela's work, Roller's work, etc.), that perform better than the numbers in Table 1. There are many other datasets that measure lexical entailment, none of which are evaluated on (apart from the original BLESS set, which is mentioned in passing). It would make sense to show that the method works on more than one dataset, and to do a thorough comparison against other work; especially given that:

4. The tone of the work appears to imply that word2vec was wrong and needs to be reinterpreted: the work leads to "unprecedented results" (not true), claims to be a completely novel method for inducing word representations (together with LSA, BOW and Word2Vec, third paragraph; not true), and suggests it has found "the best way to extract information about the semantics of a word from this model" (7th paragraph; not true). This, together with the "reinterpretation of word2vec" and the proposed "new distributional semantic models" almost makes it hard for me to take the work seriously.

---

> ### Author Response · Authors · 2017-12-22
> **No specific citations to substantiate Reviewer's claims, because there are none.**
>
> Reviewer 1 criticises the narrowness of the literature review.  This is a conference on representation learning.  This is a paper on representation learning.  Other than papers already cited, none of the work referred to is on representation learning, and none of it gives comparable empirical results.  That is why we did not cite it in the submission.
>
> Point 1: Reviewer 1 states that the model should it be easy to explain.  This is not true.  The fundamental difficulty comes from the novelty of the model, not from the writing.  If a reader is not familiar with Henderson and Popa 2016, then it is impossible to explain this vector-space model convincingly and still have space for the novel contributions of this submission.  This vector-space representation is just too different from previous vector-space representations.
>
> Point 2: There is a huge literature on word embeddings for similarity, but it is not relevant here, as explained above.  This work is on embeddings which capture entailment, in contrast to similarity.
>
> Point 3: Reviewer 1 claims "there has been a lot of work on the Weeds BLESS dataset that is not mentioned".  This is not true.  None of the authors mentioned have published results on the Weeds et al. 2014 dataset.  The only recent unsupervised results I can find for this dataset are those reported in the submission from Shwartz, which are BELOW CHANCE.  Perhaps others have tried but chosen not to publish such poor results.  We stay with this evaluation setup precisely because it has been shown that similarity-based measures do not perform well.  Unfortunately, most of the results from Shwartz et al. 2017 use a nonstandard evaluation measure which can easily be gamed, so we cannot make use of them for comparison.
>
> Point 4: Reviewer 1 provides no evidence for any of their claims in this point.  All these claims are false; we stay with the claims made in the submission.  We certainly never claim that word2vec was wrong.  Our point in this regard is that using similarity-based embeddings (like word2vec) to address entailment does not address the real issue, namely how to induce embeddings which intrinsically capture entailment.  Reviewer 1 is very confident about their judgements for someone who says themselves that they did not understand the paper.

---

### Decision · Program_Chairs · 2018-01-29
**ICLR 2018 Conference Acceptance Decision**

**Decision:**

Reject

**Comment:**

Two knowledgeable and confident reviewers suggest rejection, while one not confident reviewer suggests acceptance. I agree with the confident reviewers. All reviewers also point out that the paper is confusingly written and difficult to understand.